# First Experimental Survey of a Whole Class of Non-Commutative Quantum Gravity Models in the VIP-2 Lead Underground Experiment

Kristian Piscicchia [1,2], Antonino Marcianò [2,3,*], Andrea Addazi [2,4,*], Diana Laura Sirghi [1,2,5,*], Massimiliano Bazzi [2], Nicola Bortolotti [1,6], Mario Bragadireanu [2,5], Michael Cargnelli [2,7], Alberto Clozza [2], Luca De Paolis [2], Raffaele Del Grande [2,8], Carlo Guaraldo [2], Mihail Iliescu [2], Matthias Laubenstein [9], Simone Manti [2], Johann Marton [2,7,10], Marco Miliucci [2,†], Fabrizio Napolitano [2], Federico Nola [2,11], Alessio Porcelli [1,2], Alessandro Scordo [2], Francesco Sgaramella [2], Florin Sirghi [2,5], Oton Vazquez Doce [2], Johann Zmeskal [2,7] and Catalina Curceanu [2,5]

1   Centro Ricerche Enrico Fermi—Museo Storico della Fisica e Centro Studi e Ricerche "Enrico Fermi", Via Panisperna, 89a, 00184 Roma, Italy; kristian.piscicchia@cref.it (K.P.); nicola.bortolotti@cref.it (N.B.); alessio.porcelli@lnf.infn.it (A.P.)
2   Laboratori Nazionali di Frascati, Istituto Nazionale di Fisica Nucleare (INFN), Via Enrico Fermi 40, 00044 Roma, Italy; bazzi@lnf.infn.it (M.B.); bragadireanu.mario@lnf.infn.it (M.B.); micargnelli@gmail.com (M.C.); alberto.clozza@lnf.infn.it (A.C.); luca.depaolis@lnf.infn.it (L.D.P.); raffaele.delgrande@lnf.infn.it (R.D.G.); guaraldo@lnf.infn.it (C.G.); mihai.iliescu@lnf.infn.it (M.I.); simone.manti@lnf.infn.it (S.M.); johann.marton@tuwien.ac.at (J.M.); miliucci.marco.09@gmail.com (M.M.); napolitano.fabrizio@lnf.infn.it (F.N.); federico.nola@alumni.uniroma2.eu (F.N.); scordo@lnf.infn.it (A.S.); francesco.sgaramella@lnf.infn.it (F.S.); fsirghi@lnf.infn.it (F.S.); oton.vazquezdoce@lnf.infn.it (O.V.D.); catalina.curceanu@lnf.infn.it (C.C.)
3   Center for Field Theory and Particle Physics & Department of Physics, Fudan University, 2005 Songhu Rd., Yangpu Qu, Shanghai 200438, China
4   Center for Theoretical Physics, College of Physics Science and Technology, Sichuan University, Chengdu 610064, China
5   IFIN-HH, Institutul National pentru Fizica si Inginerie Nucleara Horia Hulubei, Strada Reactorului 30, 077125 Măgurele, Romania
6   Physics Department, "Sapienza" University of Rome, 00185 Roma, Italy
7   Stefan-Meyer-Institute for Subatomic Physics, Austrian Academy of Science, Kegelgasse 27, 1030 Vienna, Austria
8   Physik Department E62, Technische Universität München, James-Franck-Straße, 85748 Garching, Germany
9   Laboratori Nazionali del Gran Sasso, Istituto Nazionale di Fisica Nucleare (INFN), Via Giovanni Acitelli, 22, 67100 L'Aquila, Italy; matthias.laubenstein@lngs.infn.it
10  TU Wien—Atominstitut Stadionallee 2, 1020 Vienna, Austria
11  Macroarea di Scienze Matematiche Fisiche e Naturali, Università Degli Studi di Roma Tor Vergata, Via della Ricerca Scientifica, 1, 00133 Roma, Italy
*   Correspondence: marciano@fudan.edu.cn (A.M.); addazi@scu.edu.cn (A.A.); sirghi@lnf.infn.it (D.L.S.)
†   Current address: Italian Space Agency, Via del Politecnico, 00133 Roma, Italy.

**Abstract:** This study is aimed to set severe constraints on a whole class of non-commutative space-times scenarios as a class of universality for several quantum gravity models. To this end, slight violations of the Pauli exclusion principle—predicted by these models—are investigated by searching for Pauli forbidden $K_\alpha$ and $K_\beta$ transitions in lead. The selection of a high atomic number target material allows to test the energy scale of the space-time non-commutativity emergence at high atomic transition energies. As a consequence, the measurement is very sensitive to high orders in the power series expansion of the Pauli violation probability, which allows to set the first constraint to the "triply special relativity" model proposed by Kowalski-Glikman and Smolin. The characteristic energy scale of the model is bound to $\Lambda > 5.6 \cdot 10^{-9}$ Planck scales.

**Keywords:** non-commutative quantum gravity; Pauli principle violation; X-ray spectroscopy; underground experiment

## 1. Introduction

Models of non-commutative geometry represent classes of universality for several realizations of quantum gravity. Notably, non-commutativity may entail the deformation of the space-time symmetries both at the algebra and at the co-algebra levels. In particular, the latter eventuality is phenomenologically very appealing, as it is naturally connected to a possible deformation of the spin statistics theorem and, hence, to violations of the Pauli exclusion principle (PEP). Violations of the PEP, nonetheless, must be very much suppressed, as very tight constraints from the stability of ordinary matter have to be fulfilled. The strategy we adopt in this paper is related to constraining transitions that are forbidden by the PEP. The data collected by the VIP-2 lead experiment were analyzed [1,2] based on the calculated PEP-violating atomic transitions in the context of the $\theta$-Poincaré model [3]. This allowed to set the most stringent bounds on the energy scale $\Lambda$ of non-commutativity emergence in the context of $\theta$-Poincaré [4–8]. A different approach is followed in this work. A novel phenomenological investigation of the VIP-2 lead data is performed based on an analytical expansion of the PEP violation probability. By varying the power of the expansion, this allows to study a whole class of non-commutative space-time models. Hence, our analysis is extended to consider the cases of $\kappa$-Poincaré [9–12], and, for the first time, of the "triply special relativity" model proposed by Kowalski-Glikman and Smolin [13].

Within the current analysis, we exploit an analytical expansion of the PEP violation probability [4]

$$\delta^2 = c_k \left( \frac{E}{\Lambda'_k} \right)^k = \left( \frac{E}{\Lambda_k} \right)^k, \tag{1}$$

in terms of the energies involved in the atomic transitions and the energy scales $\Lambda_k$ characterizing specific models of space-time non-commutativity. $\delta^2$ accounts for the deformation of the particles' statistics, and the expansion allows to capture the behavior of several different classes of universality of quantum gravity models. We focus, in particular, on the cases corresponding to the selection of $k = 1, 2, 3$. In the phenomenological expansion, we may safely encode the order of magnitude of the theoretical estimates of $c_k$ in $\Lambda_k$ by means of the redefinition $\Lambda_k = \Lambda'_k / c_k^{1/k}$.

The $k = 1$ and $k = 2$ cases include very well-known models in the literature [4]. Indeed, these correspond to the $\kappa$-Poincaré and the $\theta$-Poincaré quantum-group-like deformation of flat space-time symmetries, respectively. The former framework was initially advocated as a realization of "doubly special relativity", a model that realizes deformation with respect to a new invariant mass scale (the Planck mass) of Einstein special relativity; the latter has been deeply connected to string theory for decades [14] through the condensation of the $B$ field, a necessary ingredient for the stability of the theory and to recover the standard model of particle physics. The case $k = 3$, which we discuss for the first time and in greater detail in this article, introduces a deformation of the space-time and momentum algebra that is appropriate for the "triply special relativity" model [13] and involves a third invariant scale (other than the velocity of light and the Planck energy), associated to the cosmological constant by the authors.

We provide an assessment of these three main relevant scenarios. In Section 2, the analytical expansion of $\delta^2$ is presented. The strategy of the analysis and setup are described in Section 3. Details of the statistical study and results are given in Sections 4 and 5.

## 2. Energy Dependence of the PEP Violation Probability in NCQG Models

We adopt the class of universality of non-commutative quantum gravity (NCQG) models, for which the quantum features of geometry are encoded in the space-time non-commutativity. This is in turn dual to a deformation of the Lorentz/Poincaré algebra that requires novel symmetry algebra structures [15,16]. The deformations of the Lie algebras of space-time symmetries force us to introduce the mathematical concepts of bi-algebra and Hopf algebra. Although accomplished with an abstract algebraic viewpoint,

the development of these latter structures entails profound physical consequences, as it affects the structure of the statistics of fermions and bosons. A strategy can then be defined that hinges on probing the microscopic structure of space-time by testing the deformation of the spin statistics relations. Specifically, tight limits on the violation of the PEP percolate into strong constraints on the observables of NCQG models. We may then distinguish, in this wide class of universality of quantum gravity models, the following sub-classes:

- For $\kappa$-Poincaré, different quantization procedures of the particle fields lead to different predictions; we will refer in particular to the following schemes: the Arzano–Marcianò (AM) procedure [10] and the Freidel–Kowalski-Glikman–Nowak (FKN) procedure [17]. In the AM quantization procedure, PEP violations are induced with a suppression $\delta^2 = E/\Lambda_1$ [10], where $E$ is the characteristic energy of the considered transition and $\Lambda_1$ is the energy scale of space-time non-commutativity. In the FKN case, the PEP violation is actually missing. In this sense, the experimental investigation of statistics violations can also provide important down-top indications on the "right" quantization procedure to solve this ambiguity in the formulation of the theory.
- The $\theta$-Poincaré model leads to the prediction (see refs. [4,18,19]) that PEP violations are induced with a suppression $\delta^2 = (E/\Lambda_2)^2$.
- Another relevant case, corresponding to the power energy expansion with $k = 3$, concerns the "triply special relativity" model, which we refer to as Kowalski-Glikman–Smolin (KS). The KS framework [13] introduces an additional infrared scale, related to the cosmological constant, and plays the role of an IR regulator. A quantum field theory endowed with the algebra of symmetries discussed in the KS framework might in principle provide IR/UV mixing, an interesting feature of some non-commutative quantum field theories. At the same time, the development of the field theoretic approach requires deepening the Hopf algebra structure of the new symmetries proposed in the KS model. Since this step is still missing at the theoretical level, our phenomenological analysis may be considered as a guidance for the theory that must be still developed for $k = 3$. Indeed, a possible interplay between the UV energy scale $\kappa$ and the IR energy scale $R^{-1}$—related to the cosmological constant $\Lambda$ by $\Lambda = R^{-2}$—may induce PEP violations at orders $k = 1$, $k = 2$ and $k = 3$. Requesting consistency for $k = 1$ and $k = 2$ with the current experimental bounds then provides strong limits on higher order corrections that can be allowed.
- For a generic NCQG model, deviations from the PEP in the commutation/anti-commutation relations can be parametrized [4] as

$$a_i a_j^\dagger - q(E) a_j^\dagger a_i = \delta_{ij} \,, \tag{2}$$

where the relevant energy scale $E$ again corresponds to the energy level difference, i.e., to the PEP-violating X-ray transition energy. Equation (2) resembles quon algebra (see, e.g., refs. [20,21]) in the case of quon fields. However, the $q$ factor does not show any energy dependence, and it is not related to any quantum gravity model. Typical deformations of particles' statistics, including the ones consistent with the $\theta$-Minkowski non-commutative space-time discussed in [1–3] and the $\kappa$-Minkowski non-commutative space-time [10], are expressed in a similar form to the quon algebra, but encode an energy dependence in the $q$ parameter, as specified in Equation (2). The q-model requires a hyper-fine tuning of the $q$ parameter. $q(E)$ is related to the PEP violation probability by

$$q(E) = -1 + 2\delta^2(E). \tag{3}$$

For a generic parametrization ($M_k$), we straightforwardly obtain:

$$M_k: \quad \delta^2(E) = \frac{E^k}{\Lambda_k^k} + O(E^{k+1}) \,, \tag{4}$$

which follows the same strategy outlined in [4] (where different values of $\Lambda_k$, the energy scales at different orders of the power series expansion, were not accounted for) and relies on the analyticity of the deformation of the spin statistics relations. The analyticity of the deformation is implied by the choice of considering theories that possess a smooth limit toward the standard undeformed theories and enables a series expansion in the power of ratios between the multi-particle states' energies and the energy scales of the new physics involved. In order to account for the possible existence of more than one deformation parameter, we consider that the energy scale at each order of the expansion might possibly be different, resulting from any possible admitted combination of energy scales and their powers. Consistently, any order one coefficient is encoded in the energy scales $\Lambda_k$. The phenomenological method includes, through the analytic expansion, the infrared limit for several different UV-complete quantum field theories. This parametrization can capture every possible first term of the power series expansions in $E^k/\Lambda_k^k$ for every possible deformation function $q(E)$ in Equation (2). In other words, constraints on $\delta(E)$ can be translated into constraints on the new physics scale(s) $\Lambda_k$ within the framework of the $M_k$ parametrization.

## 3. The Experiment and the Strategy of the Analysis

A measurement was conceived to unveil possible signals of PEP-violating atomic $K_\alpha$ and $K_\beta$ transitions in Pb. The experiment, called VIP-2 lead, was performed at the underground Gran Sasso National Laboratory (LNGS) of INFN, in which a rock overburden corresponding to a minimum thickness of 3100 m w.e. (metres water equivalent) suppresses the cosmic radiation flux by almost six orders of magnitude. The setup consists of a high purity co-axial p-type germanium detector (HPGe), about 2 kg in mass, surrounded by a 5 cm thick target of radio-pure Roman lead, shaped in three cylinders surrounding the Ge crystal—please refer to [1,2,22–24] for further details on the apparatus and the acquisition system.

Due to the third electron residing in the fundamental level, PEP-violating transitions are shifted downwards in energy with respect to standard transitions. This shift can be evaluated based on a multi-configuration Dirac–Fock and general matrix elements numerical code [25], see also ref. [26]. Table 1 reports the energies of the standard and PEP-violating $K_\alpha$ and $K_\beta$ transitions in Pb.

**Table 1.** This table summarizes the calculated values for the PEP-violating $K_\alpha$ and $K_\beta$ atomic transition energies in Pb (column labeled forbbiden). As a reference, the allowed transition energies are also quoted (allowed). Energies are in keV.

| Transitions in Pb | Allowed | Forbbiden |
|---|---|---|
| 1s - 2p$_{3/2}$ K$_{\alpha 1}$ | 74.969 | 73.713 |
| 1s - 2p$_{1/2}$ K$_{\alpha 2}$ | 72.805 | 71.652 |
| 1s - 3p$_{3/2}$ K$_{\beta 1}$ | 84.938 | 83.856 |
| 1s - 4p$_{1/2(3/2)}$ K$_{\beta 2}$ | 87.300 | 86.418 |
| 1s - 3p$_{1/2}$ K$_{\beta 3}$ | 84.450 | 83.385 |

A Bayesian analysis is performed, aimed at evaluating the upper limits $\bar{S}$ of the expected number of PEP-violating $K_\alpha$ and $K_\beta$ transitions in the target. The lead target was selected according to an accurate analysis of all the materials of the setup; hence, Pb $K_\alpha$ and $K_\beta$ atomic transitions are the only emission lines expected in the range $\Delta E = (65 - 90)$ keV, which was adopted for this study. As described in Section 2, the PEP violation probability is a function of $\Lambda_k$ and of the energy of the atomic transition under study. In order to check the sensitivity of the measured spectrum to the predicted signal as a function of the energy, a scan is performed to search for deviations from the relevant transitions (K$_{\alpha 1}$,

$K_{\alpha 2}$), ($K_{\beta 1}$, $K_{\beta 2}$, $K_{\beta 3}$) and the $K$ complex ($K_{\alpha 1}$, $K_{\alpha 2}$, $K_{\beta 1}$, $K_{\beta 2}$, $K_{\beta 3}$). Considering that the amplitudes of the standard atomic transitions are preserved, at first order, by the algebra deformation, we can neglect transitions to the 1$s$ level from levels higher than 4$p$ (see, e.g., ref. [27] for the atomic transitions intensities in Pb). In more detail, three independent analyses were performed by following the same formal procedure, in which a bin-by-bin Bayesian comparison was performed among the measured bin contents and the theoretical expectation. We will refer to the three analyses as $A_1$, $A_2$ and $A_3$, respectively. The three analyses account for a common background model—which is derived in Section 4—and an expected signal shape which only accounts for PEP violation in the $K_{\alpha}$ transitions for $A_1$, only accounts for PEP violations in the $K_{\beta}$ transitions for $A_2$ or considers PEP violations for the whole $K$ complex in the case of $A_3$. For each analysis $A_i$, and each $M_k$ parametrization, a comparison of the corresponding $\bar{S}$ with the theoretical expectation provides a limit to $\Lambda_k$.

The acquired energy spectrum, corresponding to a total acquisition time $\Delta t \approx 6.1 \cdot 10^6$ s $\approx 70$ d is shown in Figure 1.

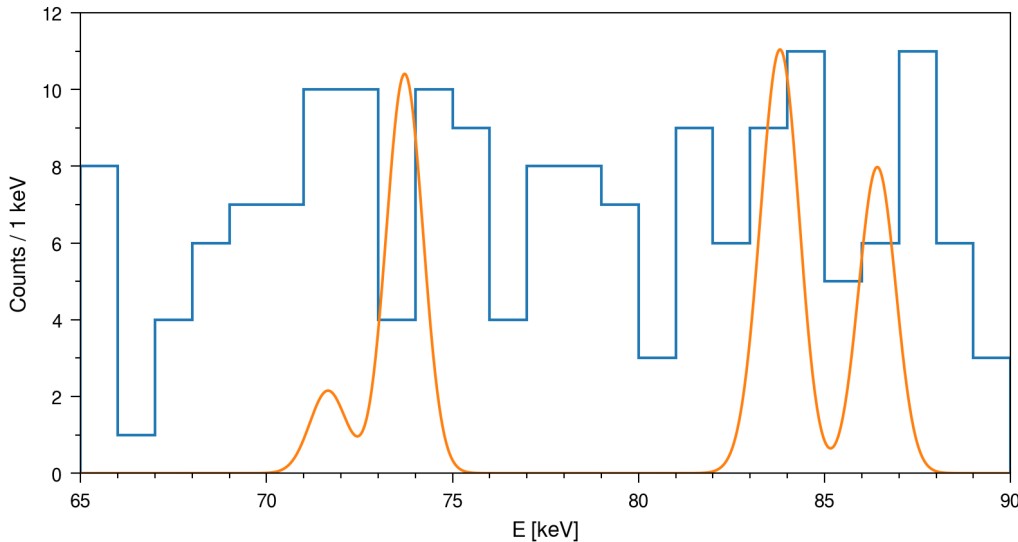

**Figure 1.** The figure shows the measured X-ray spectrum corresponding to an acquisition time of $\Delta t \approx 6.1 \cdot 10^6$ s in the region of interest. For a comparison, the expected signal distribution (with arbitrary normalization) is also shown in orange for the $A_3$ analysis and the $M_3$ parametrization.

## 4. Statistical Analysis Model

The probability density functions (*pdfs*) of $S$ are obtained based on Bayes' theorem:

$$P(S|data) = \frac{1}{\mathcal{N}} \int_0^\infty \int_{\mathcal{D}_\mathbf{p}} P(data|S,B,\mathbf{p}) \cdot f(\mathbf{p}) \cdot P_0(S) \cdot P_0(B) \, d^m\mathbf{p} \, dB, \tag{5}$$

in which $\mathcal{N}$ denotes overall normalization and the likelihood is weighted on the joint *pdf* of the experimental parameters $\mathbf{p}$ involved in the analysis, whose parameter space is indicated as $\mathcal{D}_\mathbf{p}$. With the exception of the detector resolutions (indicated by the vector $\boldsymbol{\sigma}$ and reported in Table 2 for the energies of the PEP-violating lines) and of the parameters which describe the background shape ($\boldsymbol{\alpha}$), all the other parameters, characterized by relative errors of 1% or less, are neglected; hence, $\mathbf{p} = (\boldsymbol{\alpha}, \boldsymbol{\sigma})$. The likelihood is expressed as a product of Poissonian distributions, one for each bin of Figure 1, and the expectation value of the measured number of events in the $i$-th bin is parametrized as follows:

$$\lambda_i(S,B) = B \cdot \int_{\Delta E_i} f_B(E, \boldsymbol{\alpha}) \, dE + S \cdot \int_{\Delta E_i} f_S(E, \boldsymbol{\sigma}) \, dE, \tag{6}$$

where $\Delta E_i$ is the energy range corresponding to the *i*-th bin, *B* represents the expected number of total background counts and $f_S$ and $f_B$ are the shapes of the signal and background spectra normalized to unity in $\Delta E$.

**Table 2.** Resolutions ($\sigma$) in keV estimated at the energies of the PEP-violating $K_\alpha$ and $K_\beta$ transitions.

| Transitions in Pb | $\sigma$ (keV) | Error (keV) |
|:---:|:---:|:---:|
| $K_{\alpha 1}$ | 0.492 | 0.037 |
| $K_{\alpha 2}$ | 0.491 | 0.037 |
| 1s - 3p$_{3/2}$ $K_{\beta 1}$ | 0.497 | 0.036 |
| 1s - 4p$_{1/2(3/2)}$ $K_{\beta 2}$ | 0.498 | 0.036 |
| 1s - 3p$_{1/2}$ $K_{\beta 3}$ | 0.497 | 0.036 |

The normalized signal shape is given by the superposition of Gaussians which are centered at the energies of the PEP-violating lines. Their widths are given by the experimental resolutions and their amplitudes are proportional to the rates of the corresponding PEP-violating transitions:

$$f_S(E,k) = \frac{1}{N} \cdot \sum_{K=1}^{N_K} \Gamma_K \frac{1}{\sqrt{2\pi\sigma_K^2}} \cdot e^{-\frac{(E-E_K)^2}{2\sigma_K^2}}, \qquad (7)$$

the sum extends over the number $N_K$ of PEP-violating $K$ transitions which are surveyed in each analysis $A_i$. The normalization is given by:

$$\int_{\Delta E} f_S(E)dE = 1 \Rightarrow N = \sum_{K=1}^{N_K} \Gamma_K. \qquad (8)$$

The rate of the $K_{\alpha 1}$ PEP-violating transition is:

$$\Gamma_{K_{\alpha 1}} = \frac{\delta^2(E_{K_{\alpha 1}})}{\tau_{K_{\alpha 1}}} \cdot \frac{BR_{K_{\alpha 1}}}{BR_{K_{\alpha 1}} + BR_{K_{\alpha 2}}} \cdot 6 \cdot N_{atom} \cdot \epsilon(E_{K_{\alpha 1}}), \qquad (9)$$

see ref. [2] for the derivation. In Equation (9), the ratio of branching fractions is needed to weight the relative intensity of the $2p_{3/2}$ transition with respect to the $2p_{1/2}$. $\epsilon(E_{K_{\alpha 1}})$ is the detection efficiency at the energy of the $K_{\alpha 1}$ PEP-violating transition. The efficiencies are obtained by means of Monte Carlo (MC) simulations (by using the GEANT4 software library [28]) based on a complete characterization of all the detector components [29]. Branching ratios and efficiencies are summarized in Table 3. $\tau_{K_{\alpha 1}}$ represents the lifetime of the PEP-allowed $2p_{3/2} \rightarrow 1s$ transition [30]. The rate for a generic $K$ transition is obtained by analogy with Equation (9).

**Table 3.** Detection efficiencies evaluated at the energies of the $K_\alpha$ and $K_\beta$ PEP-violating transitions and corresponding branching ratios.

| Forb. Transitions | $BR$ | $\epsilon$ |
|:---:|:---:|:---:|
| $K_{\alpha 1}$ | $0.462 \pm 0.009$ | $(5.39 \pm 0.11) \cdot 10^{-5}$ |
| $K_{\alpha 2}$ | $0.277 \pm 0.006$ | $(4.43^{+0.10}_{-0.09}) \cdot 10^{-5}$ |
| $K_{\beta 1}$ | $0.1070 \pm 0.0022$ | $(11.89 \pm 0.24) \cdot 10^{-5}$ |
| $K_{\beta 2}$ | $0.0390 \pm 0.0008$ | $(14.05^{+0.29}_{-0.28}) \cdot 10^{-5}$ |
| $K_{\beta 3}$ | $0.0559 \pm 0.0011$ | $(11.51^{+0.24}_{-0.23}) \cdot 10^{-5}$ |

According to Equation (7), the normalized signal shape depends on the specific $M_k$ parametrization through the $E_K^k$ terms which are contained in the rates. Each independent analysis $A_i$ has to be repeated accordingly for each $M_k$ in order to set constraints on the $\Lambda_k$ scale(s) of the specific model. $f_S$ does not depend on $\Lambda_k$. $f_S$ is represented (with arbitrary normalization) as an orange line in Figure 1 for the $A_3$ analysis and the $M_3$ parametrization.

In Figure 1, not even the standard atomic transitions in Pb can be disentangled due to the extreme radio-purity of the target. The background distribution is extrapolated from a maximum log-likelihood fit, which yields the normalized background shape:

$$f_B(E) = \frac{L(E)}{\int_{\Delta E} L(E)\, dE},\tag{10}$$

with $L(E) = \alpha = (3.05 \pm 0.29)\ \mathrm{counts}/(0.5\,\mathrm{keV})$, where the error accounts for both statistical and systematic uncertainties.

We chose a Gaussian background prior, for positive $B$, with a mean value of $B_0 = \langle B \rangle_G = \int_{\Delta E} L(E)\, dE$ and a standard deviation of $\sigma_B = \sqrt{B_0}$. Zero probability is imposed to negative $B$ values. A Poissonian prior was also tested to check the compatibility of the resulting $\bar{S}$ values within the uncertainty.

Considering the a priori ignorance on $S$, we opt for a uniform $P_0(S)$ distribution in the range $(0 \div S_{max})$. $S_{max} \approx 1433$ is the maximum number of PEP-violating transitions in lead, compatible with the best independent experimental bound (ref. [26]) on the PEP violation probability. $S_{max}$ is obtained from Equation (3) of ref. [26].

## 5. Results

The upper limits $\bar{S}$ are calculated for each $A_i$ analysis and for each $M_k$ parametrization by solving the equation:

$$\tilde{P}(\bar{S}) = \int_0^{\bar{S}} P(S|data)\, dS = \Pi,\tag{11}$$

for a probability of $\Pi = 0.9$. The posterior Equation (5) and the cumulative distribution functions are calculated by means of Markov chain Monte Carlo (MCMC) numerical integrations (the numerical tools are described in detail in Appendix 1 of ref. [2]). The obtained $\bar{S}$ values are affected by a relative numerical error of $\sim 2 \cdot 10^{-5}$ and are summarized in Table 4.

**Table 4.** This table summarizes the upper limits $\bar{S}$ on the expected numbers of signal counts, and the corresponding lower bounds on the scales $\Lambda_k$ for each analysis $A_i$ and for the $M_k$ parametrizations corresponding to $k = 1, 2, 3$.

| $A_i, M_k$ | $\bar{S}$ | Lower Limit on $\Lambda$ in Planck Scale Units |
|:---:|:---:|:---:|
| $A_1, k = 1$ | 11.4913 | $3.1 \cdot 10^{21}$ |
| $A_1, k = 2$ | 11.3776 | $1.4 \cdot 10^{-1}$ |
| $A_1, k = 3$ | 11.2610 | $4.9 \cdot 10^{-9}$ |
| $A_2, k = 1$ | 15.1408 | $2.8 \cdot 10^{21}$ |
| $A_2, k = 2$ | 15.1640 | $1.4 \cdot 10^{-1}$ |
| $A_2, k = 3$ | 15.1859 | $5.1 \cdot 10^{-9}$ |
| $A_3, k = 1$ | 18.7270 | $4.2 \cdot 10^{21}$ |
| $A_3, k = 2$ | 19.1847 | $1.6 \cdot 10^{-1}$ |
| $A_3, k = 3$ | 19.5993 | $5.6 \cdot 10^{-9}$ |

For each $A_i$ and $M_k$, the lower limits on $\Lambda_k$ are set by comparing the $\bar{S}$ values with the theoretical expectation for the total number of violating transitions $\mu$:

$$\mu = \sum_{K=1}^{N_K} \Gamma_K \Delta t = \frac{\aleph}{\Lambda_k^k} < \bar{S} \Rightarrow \tag{12}$$

$$\Rightarrow \Lambda_k > \left(\frac{\aleph}{\bar{S}}\right)^{1/k}. \tag{13}$$

Table 4 reports the obtained constraints on $\Lambda_k$.

## 6. Discussion

In this work, a survey was performed over the energy domain of the expected (space-time non-commutativity-induced) PEP-violating $K_\alpha$ and $K_\beta$ atomic transitions in lead. A comparison of the lower limits obtained for the non-commutativity scales $\Lambda_k$ (Table 4) identifies $A_3$, i.e., the scan over the whole $K$ complex, to be the most sensitive to the predicted energy dependence of the PEP violation probability $\delta^2$.

From the limit in Equation (13), severe constraints can be obtained on the energy scale at which space-time non-commutativity is expected to emerge in the context of a specific parametrization. Based on the analytical expansion of Equation (4), the cases of $k = 1$ (corresponding to $\kappa$-Poincaré) and $k = 2$ (corresponding to $\theta$-Poincaré) were also investigated in ref. [4]. A comparison with our results reveals that:

1. $\kappa$-Poincaré, in the AM $\kappa$-Poincaré fields' quantization model, is ruled out (we obtain $\Lambda_1 > 4.2 \cdot 10^{21}$ Planck scales).
2. $\theta$-Poincaré can be excluded up to a fraction of the Planck scale (we obtain $\Lambda_2 > 1.6 \cdot 10^{-1}$ Planck scales).

The $\theta$-Poincaré model was also examined in ref. [8], and more recently in refs. [1,2], with a different theoretical approach. The strongest bounds were obtained in the latter analyses, which excluded the $k = 2$ case for non-vanishing $\theta_{\mu\nu}$ "electric-like" components.

The feature which sets this measurement apart is the high atomic number chosen for the target material which, at the price of a limited efficiency, allows to test the space-time non-commutativity energy scale at the highest atomic transition energy (see, e.g., [31,32] for a comparison). Given the dependence on $\Lambda_k = E/\delta^{\frac{2}{k}}$, the experiment presented in this work turns out to be the most sensitive for high orders of $k$ in the parametrization in Equation (4). In particular, we performed the first measurement for the $k = 3$ case, excluding this scenario up to $\Lambda_3 > 5.6 \cdot 10^{-9}$ Planck scales.

These latter results provide valuable experimental guidance towards future developments of possible models, including the KS framework [13], in which two invariant energy scales account for the deformation of the (non-commutative) space-time symmetries.

Forthcoming analyses will deepen the phenomenology of theories accounting for the generalized uncertainty principle (GUP). These theories are indeed connected to general deformations of the symplectic symmetry structure of some non-standard theories in high energy physics. The theoretical framework hence encodes both the deformations of the Pauli exclusion principle and the generalization of the Heisenberg uncertainty principle, as was specified in [33].

**Author Contributions:** Conceptualization, K.P.; methodology, K.P., A.M., A.A., F.N. (Federico Nola) and N.B.; software, R.D.G. and F.N. (Federico Nola); validation, A.P., R.D.G. and F.N. (Fabrizio Napolitano); formal analysis, K.P., F.N. (Federico Nola), N.B., A.P., A.C., L.D.P., R.D.G., M.I., F.N. (Fabrizio Napolitano), A.S., D.L.S., F.S. (Florin Sirghi) and C.C.; investigation, K.P., A.M., A.A., M.B. (Massimiliano Bazzi), M.B. (Mario Bragadireanu), M.C., A.C., L.D.P., R.D.G., C.G., M.I., M.L., S.M., J.M., M.M., F.N. (Fabrizio Napolitano), A.S., F.S. (Francesco Sgaramella), D.L.S., F.S. (Florin Sirghi), O.V.D., J.Z. and C.C.; data curation, K.P., R.D.G., M.L. and C.C.; writing—original draft preparation, K.P. and A.M.; writing—review and editing, K.P., A.M., A.A., M.B. (Massimiliano Bazzi),

M.B. (Mario Bragadireanu), M.C., A.C., L.D.P., R.D.G., C.G., M.I., M.L., S.M., J.M., M.M., F.N. (Fabrizio Napolitano), F.N. (Federico Nola), A.P., A.S., F.S. (Francesco Sgaramella), D.L.S., F.S. (Florin Sirghi), O.V.D., J.Z. and C.C.; visualization, A.P.; supervision, K.P., A.M. and C.C.; project administration, C.C.; funding acquisition, C.C. All authors have read and agreed to the published version of the manuscript.

**Funding:** This publication was made possible through the support of the INFN institute and Centro Ricerche Enrico Fermi—Museo Storico della Fisica e Centro Studi e Ricerche "Enrico Fermi" institute. We acknowledge the support of Grant 62099 from the John Templeton Foundation. The opinions expressed in this publication are those of the authors and do not necessarily reflect the views of the John Templeton Foundation. We acknowledge support from the Foundational Questions Institute and Fetzer Franklin Fund, a donor advised fund of Silicon Valley Community Foundation (grant nos. FQXi-RFP-CPW-2008 and FQXi-MGB-2011) and from the H2020 FET TEQ (grant no. 766900). We thank the Austrian Science Foundation (FWF) which supports the VIP2 project with the grants P25529-N20, project P30635-N36 and W1252-N27 (doctoral college particles and interactions).

**Data Availability Statement:** Data available upon request.

**Acknowledgments:** We thank the Gran Sasso underground laboratory of INFN; INFN-LNGS and its Director, Ezio Previtali; the LNGS staff; and the Low Radioactivity laboratory for the experimental activities dedicated to high sensitivity tests of the Pauli exclusion principle. A.M. wishes to acknowledge support from the NSFC, through grant no. 11875113, from the Shanghai Municipality, through grant no. KBH1512299 and from Fudan University, through grant no. JJH1512105.

**Conflicts of Interest:** The authors declare no conflict of interest.

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
