# Peer review of "First Experimental Survey of a Whole Class of Non-Commutative Quantum Gravity Models in the VIP-2 Lead Underground Experiment"

_universe, doi:10.3390/universe9070321_

Round 1

Reviewer 1 Report

REFEREE REPORT FOR

Journal: Universe
Reference: universe-2422126
Title: "First experimental survey of a whole class of Non-Commutative Quantum Gravity Models in the VIP-2 Lead underground experiment"
Author(s): Kristian Piscicchia et al.
Invitation: May 22, 2023
Completed: May 24, 2023

***

The study focuses on constraining transitions that are forbidden by the Pauli exclusion principle and excludes some non-commutative spacetime scenarios accordingly.

The results of the paper are important for the community and it can be considered for publication after a revision considering the point I listed below.

[1] Equation (4) deserves a more detailed derivation as it is an important aspect of the paper. The authors should revise the paper accordingly.

Reviewer 2 Report

In the manuscript, the authors present the constraints on a class of non-commutative quantum-gravity models, including triply-special-relativity model.  However, the positioning of the manuscript seems to be ambiguous.  As a research article, it contains too many published contents in Ref.14.  As a review article, the authors just review the experimental test by VIP-2 Lead, which has been published in Ref. 14, and use “First experimental survey …” in the title.  If the authors can improve the manuscript as a research article in following aspects, I would like to recommend the manuscript publishing in Universe.

1.     In Ref. 14, the first experimental constraints have been are set with the help of PEP violation and VIP-2 Lead experiment.  Therefore, the word “First” should not appear in the title.  (“The whole class” is not a very clear and widely accepted concept, in my viewpoint. )

2.     The authors should state the quantitative constraints in Abstract.

3.     The authors should introduce some details of Ref. 14 in Introduction.

4.     The authors should emphasize the new constraints obtained in the present manuscript and clarify the difference between the new constraints and the constraints set in Ref. 14.

5.     The energies of the standard and PEP violating K_\alpha and K_\beta transitions in Pb are given in both Table 1 in the manuscript and TABLE I in ref. 14.  Why are some figures different from each other?

6.     Table 2 and 3 in the present manuscript are the same as TABLE II and III in Ref. 14, respectively.  The authors should clarify that they use the same experimental results to confine the different models or, instead, explain the necessity of listing the same tables.

7.     The shape of the expected signal distribution in Figure 1 is different from that in FIG.1 in Ref. 14.  I think that it is better to explain it more.

8.     In References, the authors listed 72 items but only cited 26 among them.  Why?

Reviewer 3 Report

The article is a continuation of two other publications by the same authors [13,14]. All works use the same experimental data obtained in the VIP-2 Lead experiment [15]. Experimental restrictions on processes that violate the Pauli principle are used to study a whole class of non-commutative  quantum gravity models. And this, of course, is important and interesting.
The article is written in a fairly understandable language and will be of interest to both theorists and experimenters. I have two minor remarks:
1. Fig. 1. I propose to indicate the time of measurements in the caption to the figure.
2. Check the spelling of links. In many references, the volume is not highlighted in black (see, for example, [1], [6-9], [13-14], [16-17, [20],…).

Round 2

Reviewer 2 Report

The authors have replied my questions and improved the manuscript.

I recommend publishing the manuscript  in the present form  in Universe.